# PGraphD*: Methods for Drift Detection and Localisation Using Deep Learning Modelling of Business Processes

**DOI:** 10.3390/e24070910

**Published:** 2022-06-30

**Authors:** Khadijah Muzzammil Hanga, Yevgeniya Kovalchuk, Mohamed Medhat Gaber

**Affiliations:** 1School of Computing and Digital Technology, Birmingham City University, Birmingham B4 7XG, UK; mohamed.gaber@bcu.ac.uk; 2Department of Computer Science, University of Reading, Reading RG6 6DH, UK; y.kovalchuk@reading.ac.uk; 3Faculty of Computer Science and Engineering, Galala University, Suez 435611, Egypt

**Keywords:** process mining, business process management, graph streams, concept drift detection, concept drift localisation, deep learning, long short-term memory

## Abstract

This paper presents a set of methods, jointly called PGraphD*, which includes two new methods (PGraphDD-QM and PGraphDD-SS) for drift detection and one new method (PGraphDL) for drift localisation in business processes. The methods are based on deep learning and graphs, with PGraphDD-QM and PGraphDD-SS employing a quality metric and a similarity score for detecting drifts, respectively. According to experimental results, PGraphDD-SS outperforms PGraphDD-QM in drift detection, achieving an accuracy score of 100% over the majority of synthetic logs and an accuracy score of 80% over a complex real-life log. Furthermore, PGraphDD-SS detects drifts with delays that are 59% shorter on average compared to the best performing state-of-the-art method.

## 1. Introduction

Businesses go through daily processes to accomplish their mission; the better their processes, the more profitable the organisation [1]. Business process management (BPM) comprises techniques and tools for identifying, analysing and monitoring business processes to optimise their performance [2]. BPM is considered as a continuous cycle comprising several phases: design, model, execute, monitor and optimise [3]. Process mining emerged from BPM as a distinct field focusing on helping organisations discover, assess and improve workflows. Process mining extracts knowledge from event data captured by various information systems to find pitfalls in an organisation. These pitfalls, if addressed, improve the organisation’s performance and productivity [4]. Since its emergence, process mining has made the BPM life cycle more effective and efficient.

The dynamics of life have a significant impact on the way business processes are being executed. Many factors such as seasonal effects, legislation changes, technological advances and unexpected events (e.g., the coronavirus pandemic) lead to changes (or drifts) in business processes over time. It can be expected that events logged over a certain period will differ from those logged over another period. This behavioural change in process execution occurring at some point in time is referred to as concept drift. Concept drift is said to occur when a change is observed in the process while it is being analysed [5,6].

Identifying concept drifts is a relevant problem in different domains, including business process analysis. Organisations always try to adapt and evolve their business processes to handle different situations. Process mining techniques are expected to consider the concept drift challenge to allow process analysis in evolving businesses [7]. In particular, the process mining manifesto [8] identifies dealing with concept drifts in process mining among the main challenges in BPM.

Concept drifts are either planned (e.g., regulatory changes) or unexpected (e.g., changes in resource capacity) [9]. Detecting unexpected drifts can be of great benefit to businesses. It can lead to extracting the actual truth on process execution, combating risks and enhancing operational processes. Concept drifts can occur from four process mining perspectives: control-flow/behavioural, organisational, case and time. They can also be divided into four types: sudden, gradual, recurring and incremental [6]. The *sudden* concept drift occurs when dynamic changes happen during process execution, i.e., a current process is substituted with a new process, and the new process takes over in subsequent process executions. This type of drift can occur as a result of a change in the law. The *gradual* concept drift occurs when two or more versions of a process co-exist, i.e., a new process exists along with an old process over a certain period of time, making it possible to execute both process versions, until the old process is gradually discontinued. For example, an organisation might introduce a new delivery process; however, the process is set to be applicable to future orders only, and all previous orders still have to follow the former delivery process. The *recurring* concept drift happens when a set of processes are changed back and forth between each other. This type of drifts are either periodic or non-periodic and are often induced by changes in the external environment, in which a business process operates. An example is sales happening in shops during certain periods of the year. The *incremental* concept drift occurs when a change is introduced incrementally into the running process until the process reaches the desired version. This class of drift is more common in organisations that follow agile business process management methodology [6].

When dealing with concept drifts in process mining, three main problems can be considered [10]: (i) drift detection (identifying whether a change has occurred), (ii) drift localisation and characterisation (identifying what has changed), and (iii) drift process discovery (discovering the evolution of the process change and how it affects the model over time). Furthermore, there are two major ways of dealing with concept drifts when analysing event logs: offline and online analysis [11]. Offline analysis refers to a scenario, where changes are discovered using historical data. In this case, the entire event log is made available to the analyst. This is appropriate for future analysis, e.g., when designing or improving processes for later deployment. Online analysis refers to a scenario, where changes need to be discovered in near-real-time. In this case, the analyst must deal with continuously incoming data or event streams. This type of analysis can be useful for organisations interested in learning behavioural changes with regards to their customers or changes in demand as they happen [6].

A number of techniques have been proposed for detecting business process drifts, e.g., [6,9,11,12,13,14,15,16]. The main objective of these techniques is to extract features, such as patterns from the process behaviour recorded in event logs and perform certain analysis to detect drifts. While some of these methods have been reported to perform well, they have a number of limitations. In particular, the majority of existing methods are designed to detect drifts that occur in event logs only (i.e., complete process execution); they do not work in online settings, where streams of events incrementally record the executions of a business process. Some of the approaches that work in online settings detect drift with a long delay as they need to wait for the trace to complete. Furthermore, since many methods rely on statistical tests over trace distributions, which may not have sufficient data samples when there is high variability in the log, they tend not to perform well on unpredictable processes whose logs contain a high number of distinct traces compared to the total number of traces.

This paper presents two new methods called PGraphDD-QM and PGraphDD-SS (where ‘P’ stands for *process*, ‘DD’ for *drift detection*, ‘QM’ for *quality metrics* and ‘SS’ for *similarity score*) for detecting sudden concept drifts in business processes from a control-flow perspective in an online scenario while considering the following two challenges: (1) change detection and (2) change localisation. The proposed methods represent business processes as graphs and thus address the task of detecting drifts in graph streams. In a graph stream, it is assumed that individual graph objects are received continuously over time. Unlike existing methods performing statistical analysis over features extracted from event logs, the proposed methods are based on deep learning. In particular, a long short-term memory (LSTM) model trained on a stream of logged events covering a previous period of time is applied to a newly generated stream of events as they occur. Graph streams representing the process behaviour of different time periods (i.e., the previous and new time periods) are generated using the decisions of the LSTM model about the most probable business process flow. According to the first proposed method, PGraphDD-QM, the model performance is then separately estimated over the previous and new graph streams using the F-score metric, and the two sets of measures are compared. The change in values of the two sets of measures is assumed to be indicative of concept drift. According to the second proposed method, PGraphDD-SS, the directly-follows graphs (DFGs) generated based on the LSTM model decisions for two different time periods are used to verify the drift, both visually by detecting structural changes and by measuring the similarity score between the adjacency matrices of the two different graphs to estimate the number of changes observed in the business process after the drift has occurred. The performance of the proposed methods is evaluated on a synthetic event log for assessing loan applications and the real-life BPIC15 dataset [17]. The latter includes five different event logs, each reflecting changes in the process of applying for a building permit performed by the Dutch municipality.

The rest of this paper is organised as follows. Section 2 discusses related work. Section 3 presents the proposed methods for detecting and localising business process drifts. Section 4 outlines the experiments performed to evaluate the proposed methods using synthetic and real-life event logs. Section 5 discusses the results of the experiments. Section 6 concludes the paper and outlines future work.

## 2. Related Work

The majority of the state-of-the-art methods for concept drift detection in business processes use the windowing technique to select traces from an event log to consider for drift analysis, alongside statistical hypothesis testing as a solution [6,9,12,14,16,18]. Some studies used clustering-based techniques to find groups of traces sharing similar characteristics that can be generalised and employed to detect drifts [11,19,20,21,22]. Other studies used graph-based analysis techniques [9,23,24] or model-to-log alignment [25,26].

Bose et al. [6] presented a method for detecting changes in business processes and identifying the regions of change. First, the authors extracted feature sets from event logs and compared their values over different windows. Then, they applied statistical hypothesis testing to investigate a significant difference between two successive windows. Martjushev et al. [16] extended the method proposed by Bose et al. [6] by using an adaptive window strategy and presented an approach to automatically detect change points by comparing significant values of two windows produced using hypothesis analysis against a predefined threshold.

Manoj Kumar et al. [15] proposed a similar method for capturing sudden concept drifts in business processes. In particular, the authors assumed that the representative appearance of feature values changes before and after the occurrence of drift. They applied a windowing strategy to select the instances for detecting and localising drifts, taking note of the sequential order of process instances in the log. The authors used statistical hypothesis testing to examine differences between successive feature values obtained using event class correlation determined by scanning the entire log, beginning with a matrix set with zero values and then updating for every next relation found while traversing the log. The look-forward window size was used to calculate each event that followed the reference event.

Maaradj et al. [14] proposed a method for detecting both sudden and gradual drifts from execution traces. The authors performed statistical hypothesis testing over the distribution of runs in two consecutive time windows. They presumed that if a drift occurred at a given time point, the distribution of runs before and after would statistically differ, and the statistical hypothesis testing could expose the difference. The authors used an adaptive window technique to automatically adjust the size of the sliding window, striking a good trade-off between accuracy and drift detection delay.

Similar to Maaradj et al. [14], Ostovar et al. [18] also used an adaptive window technique but in an online setting. Their approach involves dividing new observed events into reference and detection windows. The set of events within each window is used to build a corresponding sub-log. A contingency matrix is constructed using relations and frequencies extracted from the sub-log. The G-test of independence [27] is applied to the matrix to obtain the significance probability (*p*-value). A *p*-value below a predefined threshold suggests a drift.

Carmona et al. [12] presented an online technique for dealing with concept drifts. First, the authors applied the theory of abstract interpretation to learn an internal representation of an event log based on a polyhedron. Then, they estimated the soundness of the representation using an adaptive window technique to detect concept drifts automatically.

Li et al. [28] proposed an extensible feature that uses the sliding window technique and heuristic miner to detect and locate concept drifts in incomplete event logs. The authors further improved the Genetic Process Mining (GPM) method [29] using Weight Heuristic Miner (WHM) [30] and Differential Evolution (DE) [31] to discover the new process model of evolving processes.

Zheng et al. [11] proposed a three-stage method for detecting process drifts from event logs. First, the authors represented each trace of an event log as multiple relations such as direct succession and weak order. Then, for each relation, they inspected and partitioned the variation trend. Finally, they clustered all change points revealed by each relation to getting the final result.

Yeshchenko et al. [19] proposed a method that involves the following three steps: (i) splitting an event log into sub logs based on predefined window size and mining declarative (DECLARE) process constraints, (ii) extracting the time-series of the characteristics of the discovered constraints, (iii) clustering the series and detecting change point over them. The final step involves visualising drifts using drift maps and charts.

Seeliger et al. [9] used an adaptive window technique to split an event log into a reference window and a detection window. The authors discovered process models for both windows using the Heuristic Miner algorithm [32]. Then, they applied a statistical significance test over different graph metrics to determine the deviation of both observed process models. Using the graph metrics, the authors described changes in the process model to identify process drifts in the event log. They also performed the statistical G-test to determine whether the detection window’s process model is significantly different from that of the reference window.

Hassani et al. [13] used an adaptive window method and a modular ensemble of reasonable distance measures to detect drifts in event streams. The authors proposed the StrProMCDD algorithm that collects a batch of events in a pruning period, computes the frequency list for these events, and includes the new frequency list in a temporally ordered list used by ADWIN [33]. The window increases in size for steady process behaviour and shrinks for diverting processes, thus indicating a drift.

De Sousa et al. [22] proposed an online trace clustering approach to detect and localise drifts in online trace streams. According to this approach, traces are mapped into a vector representation used as input to a trace clustering algorithm. The resulting cluster information is used for drift detection and localisation. The authors assumed that each feature representing a significant group of traces should remain stable according to the traces’ process behaviour. Hence, to detect a drift, their method verifies whether the current value has undergone a significant variation. The method iterates over any of the clustering evolution features over time. The feature value is compared to an estimated tolerance boundary in each iteration. A value outside the tolerance boundary represents a significant behaviour shift from earlier measurements. After each detection, the drift localisation procedure is initiated, and a list of the clustering indexes where drifts have been detected is returned.

Liu Na et al. [34] presented an online framework for detecting concept drifts in event streams based on the relationship between each pair of activities in the process. The framework involves initialising the current model, which is used as a benchmark to compare with every event trace of the upcoming event streams. Next, adjacency and footprint matrices are extracted for each new trace. The matrix of the new trace is then compared with the matrix of the current model to identify differences. A metric named process model precision is calculated. A difference between the matrices indicates a drift. The method returns the activities and the difference to localise and characterise the drift, indicating if the drift is sudden or recurring.

While many of the above-reviewed methods were reported to perform well, they have limitations. For example, the windowing technique used in many concept drift detection algorithms is highly dependent on the right choice of the window size; a wrong window size can result in a high number of false negatives and false positives. Some methods, such as the one proposed in [12], are not able to locate the exact moment of a drift. The method reported in [6] is not automated; it requires human involvement in feature selection and change point recognition, making it impractical. The approach proposed in [22] is capable of dealing with drifts only presentable through trace clustering. Thus, the approach is subject to the limitations of the trace vector representation and clustering algorithms. Finally, the majority of the existing methods for detecting concept drifts in business processes are designed to work offline; i.e., they require the entire event logs featuring cases from both before and after a drift has happened. Some methods that work online detect drifts with a long delay, while others do not perform well on processes whose logs display a high number of distinct executions. Thus, detecting drifts in the online scenario (i.e., as they happen) remains a challenge.

It should be noted that drift detection in business processes is different from identifying structural changes in financial time-series data. Zarei et al. [35] define a structural change as a “disturbance that tweaks the data set substantially away from its normal path when such off-normal disturbances occur”. Several test models for structural change exist, which can be routinely applied in finance research to statistically identify structural breakpoints in financial time-series data. For example, Dong et al. [36] employed the cumulative sum of squared residuals (CUSUM) test [37] based on GARCH residuals to identify the locations of breaks or sudden changes in the volatility of gold and USD exchange rate markets. The method assumes that the given time series consists of a stable sub-series and captures the change points during market volatility. The approach involves dividing the whole time period into several sub-sample periods with different market volatilities based on the change points and analysing the relationship between gold and USD for different market periods. In general, methods developed for identifying structural changes deal with simple data structures such as numerical or categorical variables and vectors. In contrast, business processes are represented as more complex structures capturing behavioural relationships between tasks such as concurrency, conflicts and loops. As such, test models for identifying structural changes cannot be readily transferred to detecting drifts in business processes. Similarly, the methods proposed in this paper are designed to work over complex graphs rather than simple time series.

## 3. Proposed Methods: PGraphDD-QM, PGraphDD-SS and PGraphDL

### 3.1. Concept Drift in Graph Streams

Below, we introduce some basic concepts used as the basis for defining concepts related to the proposed methods in the following sections.

**Definition** **1.**
*Drift detection in business processes. Drift detection is a procedure of establishing whether a change has occurred in a process, i.e., whether the process follows a different sequence of activities in the next time period compared to that in the previous time period.*


**Definition** **2.**
*Drift localisation and characterisation. Drift localisation is a procedure of identifying the region(s) of change in a process model. A change localisation method should identify the exact point in the model where the detected drift happens, e.g., between activities A and B, without requiring a process model as input. Change characterisation involves defining the perspective of change and determining the type of drift, e.g., sudden, gradual or incremental.*


**Definition** **3.**
*Drift process discovery. Drift process discovery presents the complete change process based on drift detection, localisation and characterisation using tools that exploit and relate these discoveries. This leads to unravelling the evolution of the process change and how it affects the model over time, e.g., if a process reoccurs every season. Annotations can be used to visually demonstrate the performance of a process at different time instances, thus highlighting the process evolution.*


**Definition** **4.**
*Trace, event stream. Let A be a set of activities and A+ be a set of all nonempty finite sequences of activities from A. σ∈A+ is called a trace when σ represents a firing activity sequence of a process model. An event stream S is a multi-set of infinite event traces from A+.*


In the context of process mining, a business process can be represented as a graph built using an event stream, with nodes representing activities and edges representing the transitions between the activities. The dynamism of a business process (i.e., changes in the activities and transitions between them) can be represented as a graph stream.

**Definition** **5.**
*Graph stream. A graph stream $G_{s}$ is a sequence of elements e=(x,y;t), where x and y are node labels, and edge (x,y) happens at a time period t. A stream $G_{s}=\langle e_{1},e_{2},\ldots ,e_{m}\rangle$ typically defines a graph G=(V,E), where V denotes a set of nodes (or vertices) and E denotes a set of edges.*


The problem of detecting a concept drift in a graph stream can be formulated as locating a point *p*, when there is a difference between the observed behaviour before and after *p*. The basic idea behind detecting concept drifts is that the characteristics of the graph stream before the change point differ from the characteristics of the graph stream after the change point.

**Definition** **6.**
*Concept drift in graph streams. Let $G_{s}=\left(\sigma_{1},\sigma_{2},\ldots ,\sigma_{n}\right)$ be a graph stream, $S_{0},S_{1},\ldots \infty$ be an indefinite number of different graph streams and $T_{0}<T_{1}<\ldots <\infty$ be an indefinite number of time periods. $G_{s}\left(T_{i}\right)=G_{s_{i}}$ represents the graph being used at $T_{i}$. $S\left(T_{0}\right)=S_{0}$ is the initial graph. When a time period $T_{i}\left(0<i\leq \infty \right)$ arrives, the current graph will change into $G_{s_{i}}$ instantly, and traces will still be updated in the same event stream. Such a phenomenon is referred to as a concept drift. $T_{1},\ldots ,T_{\infty }$ are called change points. Figure 1 illustrates a concept drift occurred at change point $T_{1}$.*


Given a graph stream $G_{s}=\left(\sigma_{1},\sigma_{2},\ldots ,\sigma_{n}\right)$, the aim is thus to detect the moments, when changes happen in the stream. According to Definition 6, the model behaviour before a change is not the same as after that change. Consider the example in Figure 1. If traces from $T_{0}$ and $T_{1}$ are collected, it will be noticed they are different from each other. Therefore, the natural idea will be to compare traces before and after a candidate change point to detect a drift. Bose et al. [6], Maaradj et al. [14] and Martjushev et al. [16], all adopted this solution. However, this solution is susceptible to the following two challenges: (i) how to measure the differences between two sets of traces and (ii) how many traces to collect for testing (i.e., deciding the window size). To overcome the first challenge, these authors used feature extraction and statistical hypothesis testing, both of which are time and resource-consuming. To overcome the second challenge, they employed fixed and adaptive window size strategies, making the performance of their respective methods heavily dependent on the choice of the window size. In particular, the wrong window size or step of its adaptation can lead to false negatives and false positives, thus making it difficult to locate the exact point of a drift. This paper proposes an entirely different concept to avoid these disadvantages.

### 3.2. Proposed Approach to Concept Drift Detection

This study presents several new methods for addressing concept drifts in two ways: (i) *drift detection*—decide whether the new or recently observed process behaviour shows significant changes as compared to the previously observed process behaviour and (ii) *drift localisation*—identify the parts involved in the change of behaviour before and after the drift. Solutions for both tasks depend on the proper representation of the process behaviour in consecutive periods. The incoming event traces are represented as graph streams employing the decision of an LSTM model about the most probably next activity in the process.

In particular, the proposed drift detection and localisation methods are based on the approach introduced by the authors in [38]. According to that approach, an LSTM model is first trained on an event log as detailed in Section 3.3. The trained model is then employed to find probabilities for each event present in the log to appear in the business process next. Finally, these probabilities are used to generate a DFG as detailed in Section 3.4, representing the likely business process model as believed by the LSTM model.

### 3.3. Long Short-Term Memory for Process Mining

According to the approach proposed by the authors in [38], an LSTM model is trained to establish the most likely activity to come next in a given event sequence over time. The model’s training process was improved by broadening the context and phrasing the problem so that multiple previous time steps are considered when predicting the next time step. Specifically, event logs are pre-processed according to the following protocol and definitions.

A *NULL* label is used to mark the start of each case (or process instance). It becomes the first input activity $x_{0}$ at the current time $t_{0}$, and the target activity $y_{0}$ becomes the activity at time $t_{1}$ (this is the first activity occurring in each case of the event log). The next input activity sequence $x_{1}$ becomes the activity at the prior times $\left\{t_{0},t_{1}\right\}$, and the target $y_{1}$ becomes the next activity at time $t_{2}$. The next input activity sequence $x_{2}$ becomes the activity at the prior times $\left\{t_{0},t_{1},t_{2}\right\}$, and the target $y_{2}$ becomes the next activity at time $t_{3}$; and so on. The inputs *X* build-up for each next input sequence as prior activities join the current activity. The targets *Y* are always the activities at the next time-step $t_{n+1}$ until the last input sequence contains all the activities at prior time steps, including the current activity. At this point, the *END* label is added to mark the end of a case. This procedure is repeated until all cases in the event log are pre-processed.

**Definition** **7.**
*Predicting the next activity. Given a trace of activities $t=a_{1},a_{2},\ldots ,a_{t}$, the output of the predictive model is the next activity $\left\{a_{t+1}\right\}$.*


**Definition** **8.**
*Predicting complete traces. Given the prefix of a trace $t=a_{1},a_{2},\ldots ,a_{t}$ and **END** value to mark the end of each case, the output of the predictive model is the sequence of activities $\left\{a_{t+1},a_{t+2},\ldots ,END/a_{t}\right\}$.*


The input sequences are encoded using the *Tokenizer class* from the Keras library. The tokeniser maps each activity in an event log to a unique integer creating a sequence of integers.The prepared sequences are padded to the left using the *pad sequences()* function from Keras. This function finds the longest sequence and uses its length as a standard to pad other sequences to have the same length. The targets are dummy-encoded using the pd.get_dummies() function from the Pandas library. The function converts categorical values into dummy numerical values.

Next, a unidirectional LSTM model is defined, compiled and fitted using the pre-processed event log. The model is composed of an embedding layer (which serves as an interface between the input and LSTM layers of the network), a single LSTM layer and a fully connected dense layer as the output layer (which uses the softmax activation function to ensure the output takes the form of probability distributions). The trained LSTM model is then used to predict each next activity in the business process. A prediction probability matrix for the succeeding activity predictions is constructed. These probabilities are used to generate a visually explainable process model graph in the form of a DFG representing the decision-making process of the LSTM model about the likely business process.

### 3.4. Directly-Follows Graph for Process Mining

A DFG is a directed graph whose nodes represent activities and edges represent directly follows relations between these activities. Each edge in the DFG is annotated with a directly-follows probability, denoting the LSTM model’s next activity prediction.

**Definition** **9.**
*Directly-follows graph. Given a directly follows prediction probability matrix, its DFG is a directed graph G=(i,o,N,E), where i denotes the start event, o denotes the end event, N denotes a non-empty set of nodes and E⊆(x,y)|x,y∈N denotes a set of edges.*


**Definition** **10.**
*Directly-follows probability. Given a sequence of activities $a_{1},a_{2},\ldots ,a_{n}$ in a trace, the directly-follows probability P between $a_{1}$ and $a_{2}$ is the LSTM model’s next activity probability assignment for $a_{2}$.*


A DFG is generated based on the probabilities output by the LSTM model. In [38], a DFG is used to explain the decision-making process of the LSTM model when predicting the subsequent events in a business process. The DFG is constructed by traversing each row in the prediction probability matrix, picking the column with the highest probability, which becomes the most likely next activity, and then creating a transition between each preceding and succeeding activity by drawing an edge between the nodes. The procedure is repeated until all rows in the matrix are visited. The outcome is a process graph, which can be used to analyse the performance of the LSTM model and to identify cases that are difficult for the model to deal with so that measures can be taken to improve the model performance in such cases. A probability threshold is introduced as a parameter to allow tuning the complexity of the graph, making it possible for the level of detail in the graph to be adjusted. The graphs generated this way can also be used to perform various process mining tasks such as model discovery, conformance checking and investigating cases of non-compliance. This paper extends the work in [38] by demonstrating how the graphs constructed based on LSTM predictions can additionally be used to detect concept drifts in business processes.

### 3.5. Drift Detection: PGraphDD-QM and PGraphDD-SS

Figure 2 summarises the proposed methods for detecting concept drifts in graph streams that represent changing business processes. According to the first method (PGraphDD-QM) illustrated in Figure 2a, an LSTM model is first trained using reference traces (i.e., traces from the previous time period or stream). This LSTM model is then applied to a newly generated stream of traces. The model’s performance over the old and new streams is compared in terms of the F-score metric, which is the harmonic mean of fitness and precision. Fitness or recall is the ability of the model to reproduce the behaviour contained in the stream of traces. Precision is the ability of the model to generate only the behaviour found in the stream of traces. Intuitively, if a change was introduced to a process, an LSTM model trained on traces representing the old process (i.e., process before the change) would perform poorly on traces representing the new process (i.e., process after the change). To quantify this deterioration in performance, a threshold is introduced. The value of the F-score performance metric over the newly generated stream of events below this threshold indicates the presence of drift. In contrast, those above the threshold indicate no drift. Algorithm 1 lists the pseudo-code of PGraphDD-QM, which includes the following steps:

Split the recently observed stream of traces into two windows (lines 6–13): the detection window D (most recent traces) and the reference window R (older traces).Preprocessing (namely, encoding and padding) the traces from both the detection and reference windows as described in Section 3.3 (line 15).Define and compile an LSTM model μ and train it on the preprocessed traces from the reference window R as described in Section 3.3 and [38] to obtain fitted model $\mu^{\prime }$ (line 16).Apply LSTM model $\mu^{\prime }$ trained on the traces from the reference window R to both reference window R and detection window D to make predictions of the complete trace of each process instance (lines 18–19).Construct prediction probability matrices $P_{i}j^{R}$ and $P_{i}j^{D}$ for reference and detection windows, respectively, as described in Section 3.3 and [38] (lines 21–22).Generate two DFG process models $G^{R}$ and $G^{D}$ using the $P_{ij}^{R}$ and $P_{ij}^{D}$, respectively, as described in Section 3.4 and [38] (lines 23–24).Calculate two sets of performance metrics (fitness, precision and F-score) based on the predictions (lines 26–29).Compare metrics of the detection window with the threshold ϕ. The threshold is set based on the F-score values obtained for the reference window. An F-score below a threshold signals a drift presence (line 30).Localise the process drift by inspecting the detection window (lines 31–32).Repeat the whole process for each new run read from the stream by sliding both reference and detection windows to the right until the end of the stream is reached.

**Algorithm 1** PGraphDD-QM: concept drift detection using quality metrics
**Require:** 
Event stream: S, LSTM model: μ, Threshold: ϕ1:fScoreLog←[], DriftLog←[]2:    {read event stream, split windows, prepare data, train LSTM model and make predictions}3:

e←fetch(eventStream)

4:w←WinSize {the window size}5:

parts←splitLog(e,w)

6:**for all***i* in range(0, parts) **do**7:   j←i+18:   **if**
i=parts−1
**then**9:     j←010:    R←part(i), D←part(j)11:  **end if**12:  **while**
w≠0
**do**13:    prepare(R,D)14:    $\mu^{\prime }\leftarrow \left(\mu ,R\right)$    {train an LSTM model on the reference window}15:       {Apply the trained model to both reference and detection windows}16:    $R_{pred}\leftarrow predict\left(\mu^{\prime },R\right)$17:    $D_{pred}\leftarrow predict\left(\mu^{\prime },D\right)$18:        {build next event prediction probability matrices, construct graphs}19:    $P_{ij}^{R}\leftarrow generate\left(Pred\_Prob\_Mat,R_{pred}\right)$20:    $P_{ij}^{D}\leftarrow generate\left(Pred\_Prob\_Mat,D_{pred}\right)$21:    $G^{R}\leftarrow drawGraph\left(i,o,N,E,P_{ij}^{R},R_{pred}\right)$22:    $G^{D}\leftarrow drawGraph\left(i,o,N,E,P_{ij}^{D},D_{pred}\right)$23:        {calculate the quality metrics}24:    p←performance()25:    fitness←p.fitness()26:    precision←p.precision()27:    fScore←p.fScore(fitness,precision)28:    **if**
fScore≤ϕ
**then**29:       $print(“drift\;found\;in\;window^{”}+str\left(j\right))$30:       Report←(i,j,fScore)31:    **end if**32:  **end while**33:
**end for**



**Fitness** is the ability of a model to reproduce the behaviour contained in the log. This study employs the fitness measure proposed in [39]. It indicates the degree to which each trace in the log can be aligned with a corresponding trace produced by the process model (DFG in our case). A fitness score of 1 means that the model (DFG) can reproduce all traces in the log.

**Definition** **11.**
*Fitness. A Fitness or recall measure fitness∈L×M→[0,1] aims to quantify the fraction of observed behaviour that is allowed by the model. Let l∈L and m∈M be an event log and a process model, and T be a set of traces. Then,*

(1)
$$Fitness=\frac{\left|T(l)\cap T(m)\right|}{\left|T(l)\right|}.$$



**Precision** is the ability of a model to generate only the behaviour found in the log. This study employs the precision measure proposed in [39]. A precision score of 1 indicates that any trace produced by the process model (DFG in our case) is contained in the log.

**Definition** **12.**
*Precision. A Precision measure precision∈L×M→[0,1] quantifies the fraction of behaviour allowed by the model that was actually observed. Let l∈L and m∈M be an event log and a process model, and T be a set of traces. Then*

(2)
$$Precision=\frac{\left|T(l)\cap T(m)\right|}{\left|T(m)\right|}.$$



**F-score** is a single measure of accuracy; it is the harmonic mean of fitness and precision, calculated as
$$F-\mathrm{score}=\frac{2\times \mathrm{Fitness}\times \mathrm{Precision}}{\mathrm{Fitness}+\mathrm{Precision}}.$$

To complement PGraphDD-QM, PGraphDD-SS illustrated in Figure 2b compares the structure of the DFGs generated based on the output of the prediction by two LSTM models, one trained on traces from the reference window and the other trained on traces from the detection window covering two different periods (i.e., reference and detection). Algorithm 2 lists the pseudo-code of PGraphDD-SS, which includes the following steps:Split the recently observed stream of traces into two windows (line 5–12): the detection window D (most recent traces) and the reference window R (older traces).Preprocessing (namely, encoding and padding) the traces from both the detection and reference windows as described in Section 3.3 (line 14).Define and compile an LSTM model μ and train it first on the preprocessed traces from the reference window R as described in Section 3.3 and [38] to obtain fitted model $\mu_{1}^{\prime }$ (line 15) and then on the preprocessed traces from the detection window D to obtain fitted model $\mu_{2}^{\prime }$ (line 16).Use each of the trained LSTM models to predict the complete trace of each process instance of the window it has been trained on (lines 18–19).Construct prediction probability matrices for each of the two windows, respectively, as described in Section 3.3 and [38] (lines 21–22).Generate two DFG process models as described in Section 3.4 and [38] using the respective prediction probability matrices of each window (lines 23–24).Build adjacency matrices for each of the constructed DFG process models, respectively (lines 26–27).

**Definition** **13.**
*Adjacency matrix. Let G be a DFG with a vertex set $V=v_{1},\ldots ,v_{n}$. G can be transformed into an adjacency matrix A, where A is a square n×n matrix, such that its element $A_{ij}=1$ when there is an edge from vertex $v_{i}$ to vertex $v_{j}$, and $A_{ij}=0$ when there is no edge.*


8.Compare the two DFG process models by calculating the similarity score using the generated adjacency matrices (lines 28–34). Compare the similarity score to the threshold set based on the highest similarity score obtained.9.Analyse changes and localise the position of drift (lines 35–39).10.Repeat the whole process for each new run read from the stream by sliding both reference and detection windows to the right until the end of the stream is reached.

**Calculating similarity score.** In addition to visually comparing the similarity between the two graphs representing two snapshots of a business process taken over different windows, the similarity is also verified formally in PGraphDD-SS. To achieve this, the two graphs are first converted into two adjacency matrices. Then, the similarity score between the matrices is calculated by taking the sum of the differences between the values in corresponding cells of the two matrices and dividing it by the number of non-zero values in the matrix with the least number of activities. If the two graphs, namely, the first graph representing the reference window and the second graph representing the detection window, happen to differ in size (i.e., one has more transitions than the other), the transitions present in one window but absent in the other are introduced when constructing the adjacency matrix of the latter to enable the calculation of the similarity score. These additional transitions are initialised to zeros since they have not occurred in reality. A similarity score of 1 indicates that the two graphs are identical (i.e., there is no drift), whereas a similarity score of 0 indicates that they are entirely different (i.e., represent different business processes). A similarity score between 0 and 1 indicates drift presence. For the calculation of the similarity score, binary matrices are constructed. A value of 1 in a cell indicates that the transition between the corresponding two activities exists, whereas a value of 0 represents the fact that there is no transition between the two activities.
**Algorithm 2** PGraphDD-SS: concept drift detection using similarity score**Require:** 
Event stream: S, LSTM model: μ, Threshold: ϕ1:simScore←[], DriftLog←[]2:e←fetch(eventStream)3:w←winSize {the window size}4:parts←splitLog(e,w)5:**for all***i* in range(0, parts) **do**6:  j←i+17:  **if**
i=parts−1
**then**8:    j←09:    R←part(i), D←part(j)10:  **end if**11:  **while**
w≠0
**do**12:    prepare(R,D)    {compile and train two lstm model on the reference and detection window}13:    $\mu_{1}^{\prime }\leftarrow \left(\mu ,R\right)$, $\mu_{2}^{\prime }\leftarrow \left(\mu ,D\right)$14:        {Make predictions for both}15:    $R_{pred}\leftarrow predict\left(\mu_{1}^{\prime },R\right)$16:    $D_{pred}\leftarrow predict\left(\mu_{2}^{\prime },D\right)$17:        {build next event prediction probability matrices, construct graphs}18:    $P_{ij}^{R}\leftarrow generate\left(Pred\_Prob\_Mat,R_{pred}\right)$19:    $P_{ij}^{D}\leftarrow generate\left(Pred\_Prob\_Mat,D_{pred}\right)$20:    $G^{R}\leftarrow drawGraph\left(i,o,N,E,P_{ij}^{R},R_{pred}\right)$21:    $G^{D}\leftarrow drawGraph\left(i,o,N,E,P_{ij}^{D},D_{pred}\right)$22:        {generate adjacency matrices, measure similarity score}23:    $A_{ij}^{R}\leftarrow generate\left(adjacency\_matrix,G^{R}\right)$24:    $A_{ij}^{D}\leftarrow generate\left(adjacency\_matrix,G^{D}\right)$25:    p←performance()26:    simScore←p.getScore()27:    M←0 {set a counter of non-zero values}28:    $M\leftarrow getM(A_{ij}^{R},A_{ij}^{D})$29:    $absoluteVal\leftarrow (A_{ij}^{R}-A_{ij}^{D})$30:    sumVal←absoluteVal.abs().sum().sum()31:    simScore←1−sumVal/M32:    **if**
simScore≤ϕ
**then**33:       $print(‘‘drift\;found\;in\;window^{\prime \prime }+str\left(j\right))$34:          {change point detected and reported}35:    **end if**36:    Report←(i,j,simScore)37:**end while**38:**end for**

### 3.6. Change Localisation: PGraphDL

Detecting a drift without localising it does not provide a complete picture of the change that occurred in a process. Change localisation aims to identify the region of the drift and unravel what has changed in the behaviour of a process. While drift detection alerts organisations that a process has changed, drift localisation sheds more light on where the process has changed.

This study proposes to use the process graphs constructed based on two different windows (reference and detection) as described in Section 3 to identify the locations of process drifts. The proposed change localisation method, called PGraphDL, allows gathering details about the structural change and modifications detected when comparing two process graphs.

According to the proposed change localisation method PGraphDL, two process graphs (*graph A* and *graph B*) constructed based on two different windows (reference and detection), respectively, are taken as input (Figure 3). A user selects any path of interest from the base process model by specifying an index. The best matching path is searched in graph B for each possible path in graph A by computing the positional score for each candidate path in graph B. The positional score is calculated as the number of activities in graph A located in the same position in graph B divided by the length of the selected path in graph A. For example, if there are five activities in the selected path of graph A and four matching activities in graph B, then the positional score is 4/5. The candidate path in graph B with the maximum positional score is selected as the best matching path in graph A. A maximum positional score of 1 indicates that the paths in two graphs are identical (i.e., there is no drift). At the same time, each activity in graph B that is not found in the same position as the activity in question in graph A is declared as positional drift. For example, if ‘b’ is an activity in the selected path of graph A with a positional index of ‘1’, but there is an activity ‘k’ in the positional index ‘1’ of graph B, then ‘k’ is declared as positional drift. In the experiments outlined in the next section, the probability threshold was set to 0 when identifying candidate paths (i.e., all transitions in the graph inclusive). The process graph of each path is constructed with drift positions highlighted in dotted circles to visualise drifts. The graph of the selected path is also displayed to make the explanation clearer (Figure 4).

## 4. Experiments

All the stages of PGraphDD-QM and PGraphDD-SS illustrated in Figure 2 were implemented as a set of Python scripts using Python 3.6. The LSTM models were built using the Keras [40] and Tensorflow [41] libraries. The process graphs were generated using the Graphviz library [42]. The experiments were carried out using the Google Colab free Tesla K80 GPU.

Two publicly available event logs were used to evaluate the performance of the proposed concept drift detection methods, namely, the loan application process [43] and Dutch municipality (BPIC 2015) [17] logs. The details of the two datasets are presented below in turn. Accuracy (calculated as a harmonic mean of precision and recall) and mean delay (calculated as the average number across all windows of log traces between the point when a drift happened and when it was detected) were used as performance metrics [44]. The results obtained for the loan application process logs [43] were compared to those reported in [9,14,22]. While there exists a study using the BPIC 2015 logs for drift detection [13], the results reported cannot be directly compared to ours.

### 4.1. Loan Application Process Dataset

The loan application process dataset consists of 72 synthetic event logs generated from a base model comprising 15 activities, one start and three end events. The logs exhibit different control-flow structures, including loops, parallel and alternative branches (Figure 5). To generate the logs, the base model was modified systematically by applying, in turn, one out of twelve simple change patterns described in [45] (Table 1). These modifications reveal different change patterns, which are categorised into insertion (“I”), resequentialisation (“R”) and optionalisation (“O”). More complex drifts were created by combining the simple change patterns; this involved randomly applying a pattern from each category in a nested way, thus resulting in additional event logs: “IOR”, “IRO”, “OIR”, “ORI”, “RIO” and “ROI”.

To vary the distance between the drifts, four event logs of sizes 2500, 5000, 7500 and 10,000 were generated for each of the change patterns by combining a fixed number of alternating instances from the base model, then a fixed number of instances from the modified model, leading to a total of 72 logs. Each event log generated in this way contains precisely nine process drifts.

### 4.2. BPIC 2015 Dataset

BPIC 2015 is a real-life dataset, which includes event logs provided by each one of the five Dutch municipalities [17]. The logs contain many activities, each labelled with a code and a Dutch and English label. While the processes across the five municipalities should be identical, they differ in reality. A behaviour in each municipality is not observed in each of the other municipalities. There are distinctions in sub-processes between the municipalities, both in the frequency and behaviour. The differences may have resulted from the changes made to procedures, rules or regulations. As there are about 1170; 828; 1349; 1049 and 1153 different execution paths for the BPIC2015_1; BPIC2015_2; BPIC2015_3; BPIC2015_4 and BPIC2015_5 logs, respectively, almost all cases are unique from the control-flow perspective. Similar to [13], for the experiments presented in this study, the five logs were merged together to get one log with four reliable concept drifts.

### 4.3. Experimental Setup

The two proposed drift detection methods were tested in two experiments: (1) drift introduced at the start of the detection window (Figure 6) and (2) drift introduced in the middle of the detection window (Figure 7). In the first experiment, all traces in the detection window were generated by one process model that was different from that used to generate traces for the reference window. In the second experiment, the detection window contained traces generated by two different process models: an old one used for the reference window and a new one. The point of the switch from the old to the new process in the detection window (i.e., the ratio between the number of traces generated by the old and new process models) was varied in the second experiment to assess the sensitivity of the proposed drift detection methods to the number of traces generated by the old process model still present in the analysed window containing a drift.

To clearly demonstrate the ability of the proposed methods to detect drifts in the first experiment, the LSTM model was initially trained on the first half of the reference window and applied to its second half. In this case, the model is expected to achieve an F-score close to 1 since the same process model generated traces in both portions. This F-score value was used to set the threshold for PGraphDD-QM. Next, the model was trained on the second half of the reference window and applied to the first half of the detection window. Since the model was trained on traces generated by one process model but applied to traces generated by another process model, a drop in the F-score value below the threshold is expected, indicating a drift. The procedure was repeated by shifting windows forward until reaching the end of the considered log. For all the iterations, it was noted whether the F-score value was always above the threshold when the LSTM model was trained and applied to traces generated by the same process model (i.e., not triggering false alarms) and below the threshold when the LSTM model was trained on traces generated by one process model but applied to traces generated by another process model (i.e., detecting drifts when they actually happened). For PGraphDD-SS, the similarity score was compared across the windows in the same manner, assuming that a similarity score above the threshold indicated no drift and a similarity score below the threshold meant a drift.

The second experiment was designed to explore the behaviour of the proposed drift detection methods on the stitches of the traces generated by different process models and obtain the delay metric. In this case, each detection window was constructed to include traces before and after the change point (i.e., drift) at different percentages. Initially, the detection window was set to have 90% of traces generated by the old process model and 10% of traces generated by the new process model. The F-score value was checked, and a drift was assumed to be detected if the F-score value was below the set threshold. If drift was not detected, the detection window was modified to include old and new traces at the ratios of 80%:20%, 70%:30% and so on until drift was detected. The delay was set to the number of traces generated by the new process model at the point of the detected drift (i.e., if the drift was detected at a ratio of 30%:70%, the delay was set to the number of traces in the 30% block). A short delay between a change and its detection is highly desirable.

The following experimental protocol was adopted to evaluate the proposed methods for detecting concept drifts in business processes:For both methods, the threshold was tuned for each dataset, according to the values obtained over the very first reference windows (which contain no drift by default).For the embedding layer of the LSTM model, the vocabulary size was set to the number of unique activities in the considered log. The vocabulary size guided the number of embedding dimensions; hence it varied across the event logs.The network architecture of the LSTM model included one hidden LSTM layer with a dropout probability of 0.2. The dropout technique was used to avoid over-fitting and improve learning by randomly removing a cell from the network.The number of training epochs was set to 50 for all the synthetic logs and 10 for the BPIC 2015 logs based on preliminary experimentation.The parameters of the LSTM model, namely, the number of embedding dimensions, batch size and LSTM neurons, were tuned for the different event log sizes for better performance. In particular, 250; 500; 750; 1000 embedding dimensions and batch sizes were used for each of 2500; 5000; 7500; 10,000 loan application process logs, respectively; 1000 embedding dimensions and a batch size of 1000 were used for the BPIC 2015 logs; 100 neurons were used in the LSTM layer for all the logs.

## 5. Results

### 5.1. Drift Detection: Loan Application Process

In the first experiment for the loan application process dataset, the thresholds were set to 0.9 and 1.0 for PGraphDD-QM and PGraphDD-SS, respectively. PGraphDD-QM achieved an average accuracy score of 99% across all log sizes over 11 out of 17 change patterns. For the remaining change patterns (’cb’, ‘cm’, ‘ior’, ‘pl’, ‘rio’ and ‘lp’), the method achieved an average accuracy of 66%. These results suggest that the effectiveness of PGraphDD-QM depends on the nature of the event log. For example, in the case of the ‘cb’ change pattern, the fragment “‘Prepare acceptance pack’—‘Check application form completeness’” always present in the reference window may be omitted in the detection window (i.e., the process branches, meaning that the two activities are executed in some cases but not always, resulting in a portion of traces containing the two activities and in a portion of traces not containing them). Since the LSTM model trained on the reference window, where the activities are always present, would still get points for correctly predicting the case of these activities being present in the detection window, PGraphDD-QM may fail to detect the drift if the proportion of the traces containing the activities is greater than the proportion of the traces not containing these activities.

PGraphDD-SS achieved an average accuracy score of 98% over 16 out of 17 change patterns in the first experiment. However, over the remaining ‘ior’ change pattern, PGraphDD-SS achieved a low average accuracy score of 58%, which can be explained by the composite nature of this change pattern, where simple change patterns are nested within each other. Such a build means that there is a point in time when a new behaviour resembles an old behaviour from a previous time, which is challenging for the method to handle as indicated by the four false negatives made by the LSTM model for each of the log sizes.

Table 2 lists F-scores and Table 3 lists similarity scores achieved by PGraphDD-QM and PGraphDD-SS, respectively, over each pair of windows slid across the ‘cd’ log of sizes 2500, 5000, 7500 and 10,000 as an example to explain how the decision on whether a drift have occurred or not is derived. F-score and similarity values above the set thresholds indicate drift absence. In contrast, a drop in the values below the thresholds indicates the presence of a drift. Considering our knowledge of the event log, we can confirm whether drift happened or not in reality. The Yes/No answers in the *‘Drift detected?’* column in both tables indicate whether the methods detected a drift or not, respectively. The Yes/No answers in the *‘Actual drift?’* column indicate whether the drift has actually occurred or not, respectively. The Yes/No answers in the *‘Correct detection?’* column indicate whether the methods detected the drift correctly or not, respectively. It can be observed that the decisions reported in the *‘Detected drift?’* column of both tables are confirmed as accurate detections in the *‘Correct detection?’* column. This means that both methods were able to detect all nine drifts for the ‘cd’ logs of all sizes. Indeed, it can be noticed from the tables that high F-score and similarity values were returned for each window pair containing traces generated by the same process model and low values for each window pair containing traces generated by different process models.

Further analysis of the results revealed that PGraphDD-SS is more stable and accurate on average than PGraphDD-QM (Figure 8). At the same time, it should be noted that PGraphDD-SS requires training two LSTM models, whereas PGraphDD-QM—only one. Hence, PGraphDD-QM can be recommended for resource-constrained environments, while PGraphDD-SS should be used when achieving accurate and stable results is more critical than efficiency.

### 5.2. Drift Detection: BPIC 2015

Table 4 lists F-scores obtained for BPIC 2015 using PGraphDD-QM, with the threshold F-score value set to 0.9. It can be noticed from the table that PGraphDD-QM was able to find all four concept drifts. However, it reported two false positives, thus achieving an accuracy score of 80%. The false positives can be attributed to the complex nature of the five business processes and the high variability of traces within each of the five logs, as mentioned in Section 4.2.

Table 5 reports results achieved by PGraphDD-SS over the BPIC 2015 dataset. It can be noticed from the table that significantly lower similarity scores were achieved across all window pairs of the BPIC 2015 dataset compared to the Loan Application Process dataset, which again can be explained by the much more complex nature of the BPIC 2015 logs and higher variability across traces within each log compared to the Loan Application Process logs. The threshold similarity value for BPIC 2015 was set to 0.3. Three drifts were correctly detected, while one was missed, resulting in an accuracy score of 75%.

The results obtained for BPIC 2015 indicate that both PGraphDD-QM and PGraphDD-SS can perform well regardless of the complexity and peculiarities of logs but require setting F-score and similarity threshold values on case by case basis. It is important to note that the thresholds can be set for each case automatically based on the results obtained over two windows known covering traces generated by the same process model.

### 5.3. Drift Localisation

In addition to detecting drifts, it is important to be able to localise them, i.e., correctly identify the activities/fragments that changed, e.g., added or removed. To achieve that, we visually and analytically (i.e., using the proposed PGraphDL method introduced in Section 3.6) compare graphs generated over two different windows. The correctness of identified changes for the Loan Application Process dataset can be validated using the knowledge about the changes injected into the event logs (Section 4.1).

As an example of visual drift localisation using PGraphDL, Figure 9 shows two directly follows process graphs constructed for a reference window before a drift and a detection window after the drift, respectively. The process graphs were generated using the Graphviz library [42] according to the procedure described in Section 3.4. The reference window graph (Figure 9a) represents the base BPMN model of the loan application process (Section 4.1, Figure 5), whereas the detection window graph (Figure 9b) represents a modified version of the base model after introducing the ‘sw’ change pattern. It can be observed that before the drift (Figure 9a), activities ‘Prepare acceptance pack’ and ‘Check if home insurance quote is requested’ happened before activity ‘Verify repayment agreement’. However, after the drift (Figure 9b), activities ‘Prepare acceptance pack’ and ‘Check if home insurance quote is requested’ swapped place with activity ‘Verify repayment agreement’.

Table 6 lists the reasons extracted from the pair of graphs generated for some of the change patterns using PGraphDL. A reason is marked correct if the modification to the event log corresponds to the activities identified as the reasons for the process drift. In all cases, the correct reason for the process drift from the graphs are extracted.

To evaluate PGraphDL, a path selected from the graph representing the base loan application process was compared to the paths from the graphs representing the modified versions of the base process. After selecting the candidate path for each graph based on the positional scores, drift positions were visualised as described in Section 3.6.

Figure 10 shows an example of localising drifts in the ‘sw’ log, where two fragments are swapped in some specific parts of the log. It can be noticed from the figure that before the drift (path A), activity ‘Check if home insurance quote is requested’ is executed after activity ‘Prepare acceptance pack’, whereas after the drift (path B), activities ‘Prepare acceptance pack’ and ‘Check if home insurance quote is requested’ swapped place with activities ‘Verify repayment agreement’ and ‘Send home insurance quote’. Figure 11 considers the ‘re’ log, where a fragment is added or removed in some specific parts of the log. It can be noticed from the figure that before the drift (path A), activity ‘Appraise property’ is followed by activity ‘Assess eligibility’, then by activity ‘Reject application’, whereas after the drift (path B), activity ‘Assess eligibility’ is removed, while activity ‘Reject application’ happens after activity ‘Appraise property’. Finally, Figure 12 considers the ‘rp’ log with a substituted activity. It can be noticed from the figure that before the drift (path A), activity ‘Verify repayment agreement’ is executed after activity ‘Send home insurance quote’, whereas after the drift (path B), the activity ‘Verify repayment agreement’ is substituted with activity ‘Replaced activity’.

Due to limited space, only one example of drift localisation is presented for BPIC 2015. In Figure 13, one path from the graph generated using the BPIC2015_1 reference window is compared to the closest path detected by the proposed drift localisation method in the graph generated using the BPIC2015_2 reference window. It can be noticed from the figure that before the drift (path A), 46 activities were executed sequentially. After the drift (path B), some of those activities were replaced, while other, new activities were introduced. For example, after the drift (path B), activities ‘01_HOOFD_015’ and ‘01_HOOFD_030_2’ in the reference graph, were replaced with activities ‘05_EIND_010’ and ‘03_GBH_005’ in the detection graph. It can also be noticed that after the drift (path B), some activities such as ‘01_HOOFD_061’ and ‘01_HOOFD_510_2’ executed in path A are seen to happen sporadically. Drifts can be easily localised in the same manner for the other pairs of the Loan Application Process and BPIC 2015 logs.

### 5.4. Comparison with State-of-the-Art Methods

Table 7 lists accuracy scores for each change pattern averaged over the four event log sizes as achieved by the proposed PGraphDD-QM and PGraphDD-SS methods, and existing approaches reported in [9,14,22]. The highest accuracy values for each change pattern across all the compared methods are highlighted in bold.

PGraphDD-QM was able to detect drifts with high accuracy between 90% and 100% in 12 out of 17 change patterns, while PGraphDD-SS—in 16 out of 17 change patterns (Table 7). PGraphDD-QM outperformed the state-of-the-art methods over some change patterns such as ‘cd’, ‘cf’ and ‘rio’. However, it did not perform on par with the other methods over change pattern ‘ior’. PGraphDD-SS outperformed PGraphDD-QM (Table 7) over change patterns ‘cb’, ‘cm’, ‘lp’, and ‘rio’. For example, PGraphDD-QM could not find all the drifts in “cb” because it may have been tricked by the appearance of both skippable and non-skippable fragments in the detection window. However, according to PGraphDD-SS, these changes were captured in the adjacency matrices generated from the pair of graphs and thus could not be missed in the similarity score calculation. Overall, the proposed methods match the performance of the state-of-the-art methods while not requiring generating features manually.

For a fair comparison with Maaradji et al. [14] and Seeliger et al. [9], a fixed window size of 100 traces was used for all log sizes in the second experiment assessing the delay of the proposed methods in detecting drifts. Figure 14 and Figure 15 show the mean delay per change pattern for each of the four log sizes for PGraphDD-QM and PGraphDD-SS, respectively. It can be noticed from the figures that PGraphDD-QM performs better on reasonably small log sizes (5000), whereas PGraphDD-SS performs better on larger log sizes (7500 and 10,000). The performance of both methods on very small log sizes (2500) is not stable.

Figure 16 compares the mean delays per change pattern across the four log sizes achieved by PGraphDD-QM and PGraphDD-SS with those achieved by the methods reported by Maaradji et al. [14] and Seeliger et al. [9]. It can be observed from the figure that both PGraphDD-QM and PGraphDD-SS outperform the state-of-the-art methods, with PGraphDD-SS achieving the most stable results and the shortest drift detection delays compared to the other three methods.

Table 8 lists mean delays for each change pattern averaged over the four different event log sizes for PGraphDD-QM and PGraphDD-SS, as well as state-of-the-art methods reported in [9,14,22]. The lowest mean delay values for each change pattern are highlighted in bold. It can be noticed from the table that PGraphDD-SS performs better than the other methods in 12 out of 17 logs (cb, cd, cf, cm, cp, iro, lp, oir, pl, re, rio, sw).

We found only one study conducted over BPIC 2015 logs [13]. However, the metrics employed in that study are not comparable to those reported in this and other studies on drift detection in process mining. In particular, the study presented in [13] used measures such as dependency, edge and routing distances to identify drifts. The authors reported that out of the four drifts resulting from concatenating the five BPIC 2015 logs, the first three drifts were detected but not the fourth one when using the distance measure. The edge distance allowed to detect all four drifts but with low confidence for the first and last drift. Finally, the routing distance also allowed to detect all drifts but with low confidence for the first drift. As discussed in Section 5.2, PGraphDD-QM was able to detect three out of four drifts, while PGraphDD-SS was able to detect all four drifts. However, the overall performance of the two methods was affected by a number of false positives reported over pairs of windows taken from the same logs (something not explored in [13]). At the same time, it is not possible to provide a full evaluation of false positives due to the absence of the ground truth about the individual municipality logs (i.e., there could be changes in the business processes of each municipality over time constituting drifts that are not formally recognised).

## 6. Discussion

The proposed methods can find use in many different application domains. For example, the adoption of the methods in healthcare may provide efficient solutions for saving lives. In particular, the proposed methods can be employed in discovering disease trajectories, which can reveal disease correlations and temporal disease progression, thus equipping clinicians with tools for predicting and preventing future complications in individual patients [46]. Previous studies have based their solutions for discovering disease trajectories on statistical analysis [47] and knowledge graphs [48]. Both approaches have their limitations: while the former approach is prone to statistical bias, the latter is not scalable and requires significant expert input. Furthermore, both approaches are not designed to track changes in disease trajectories over time, e.g., to study the implications of the COVID-19 pandemic on population health and determine the subset of comorbidities contributing to the death outcome following the coronavirus disease. In contrast, the proposed methods can be easily applied to temporal sequences of diagnoses extracted from electronic healthcare records at scale and in a continuous manner, allowing clinicians to track changes in disease progression patterns, perform temporal analysis of comorbidities using the generated graphs and predict patient outcomes based on their history.

The oil and gas industry is another complex domain, where the proposed methods can be useful for preventing theft, improving the efficiency of supply chain management and adapting to world events in a timely manner. The industry involves many interconnected tasks and parties [49]. The actions of every party can be logged and cases of noncompliance identified easily by visualising the intended and actual processes as graphs. Any changes and weaknesses in the supply chain management can be equally efficiently spotted through detecting and localising drifts in the discovered process graphs evolving over time.

It should be noted that the proposed methods are designed to detect changes in business processes rather than predict process outcomes. For example, the methods can detect changes in the process required from bank staff to follow to approve a loan, while they are not suitable for predicting whether a specific loan application will be approved or not.

To make it easy for different companies to use the proposed methods in their practice, we intend to package the proposed methods as a plugin to established open-source process mining software such as ProM.

## 7. Conclusions and Future Work

Modern-day business processes are prone to changes over time due to changing circumstances and conditions. To proactively respond to these changes, also known as concept drifts, businesses require mechanisms to detect and analyse them. This paper presented three novel methods (PGraphDD-QM, PGraphDD-SS and PGraphDL) for detecting and localising sudden concept drifts in dynamic business processes based on streams of logged events. The first drift detection method involves training an LSTM model on a previous stream of events and applying it to a newly generated stream of events. The performance of the model over the old and new streams is compared based on the F-score metric. Similar F-score values over the two streams indicate no drift, whereas a drop of the F-score value over the new stream below a set threshold is indicative of a drift. The more significant drop in the F-score value, the more changes are expected to be present in the new stream of events compared to the old one. The second drift detection method involves training two LSTM models, each on a stream of events covering two different periods, respectively. These models are then used to generate two graphs representing the business processes for the two analysed periods as believed by the LSTM models. Next, two adjacency matrices are generated based on the two graphs to measure the similarity between the two business processes. A drop in the similarity score below a set threshold is indicative of a drift. The more significant drop in the similarity score, the more changes are expected to be present in the new stream of events compared to the old one. Finally, the paper detailed how the graphs representing two different periods of a business process can be used to visually and analytically localise the parts of the process that have changed.

An evaluation of the proposed methods using synthetic and real-life logs demonstrated that the methods perform on par with state-of-the-art methods, achieving similar accuracy in detecting drifts, with shorter delays compared to the other methods, while also offering the following advantages over existing solutions. First, unlike the method based on a statistical analysis of graphs representing business processes, the proposed methods employ deep learning, which does not require the user to construct features. Second, the proposed methods employ graphs that explain the decision-making process of the deep learning models predicting the next activities in the sequence of business processes, thus allowing the user to verify what has changed in the process when a drift is detected. Finally, the relative insensitivity of the LSTM model to interval length may have contributed to detecting drifts with minimal delay.

In the future, we plan to evaluate the proposed methods on several other real-life event logs, explore the ways of applying the methods to analysing chains of events affecting the stock market and adapt the methods to enable them to detect gradual drifts. Furthermore, we intend to extend the drift localisation method by enumerating all the detection paths in descending order of proximity to the reference path. Another avenue for future work is to package the proposed methods as a library for existing popular process mining software and provide the user with an interactive visualisation tool for extensive exploration of changes in case they are more significant than those considered in this study.

## Figures and Tables

**Figure 1 entropy-24-00910-f001:**
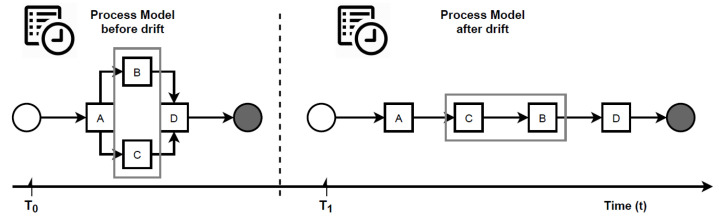
Concept drift phenomenon in process mining.

**Figure 2 entropy-24-00910-f002:**
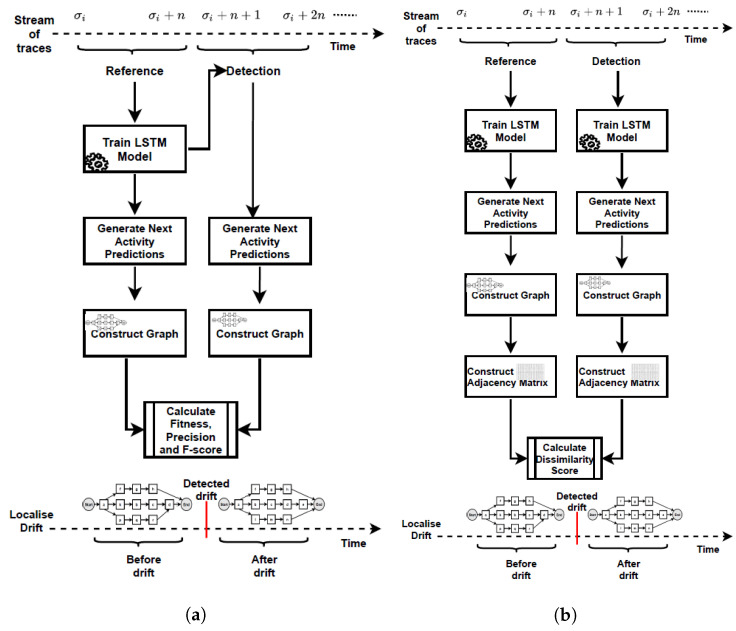
Proposed approach to detecting and localising concept drifts in business processes: (**a**) PGraphDD-QM; (**b**) PGraphDD-SS.

**Figure 3 entropy-24-00910-f003:**
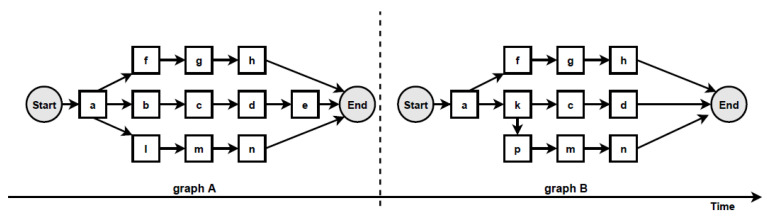
Two process graphs constructed based on two different windows: reference and detection.

**Figure 4 entropy-24-00910-f004:**
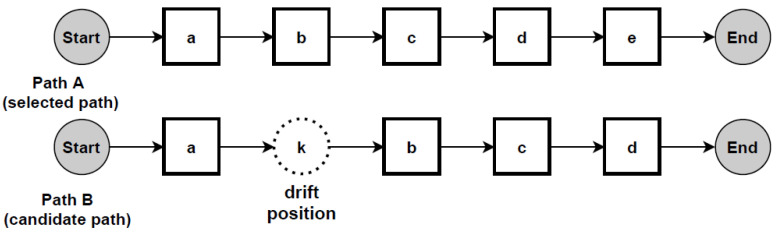
Change localisation according to the proposed PGraphDL method: Path A is the selected path from graph A, while Path B is the candidate path from graph B with drift position highlighted.

**Figure 5 entropy-24-00910-f005:**
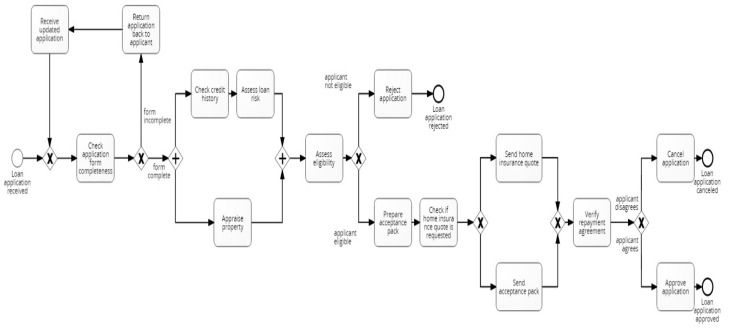
Base BPMN model of the loan application process.

**Figure 6 entropy-24-00910-f006:**
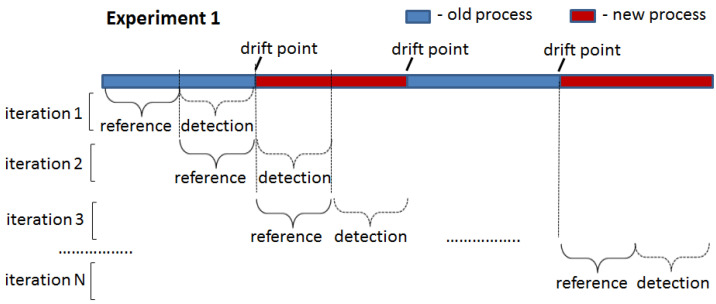
Experiment 1: drift introduced at the start of the detection window.

**Figure 7 entropy-24-00910-f007:**
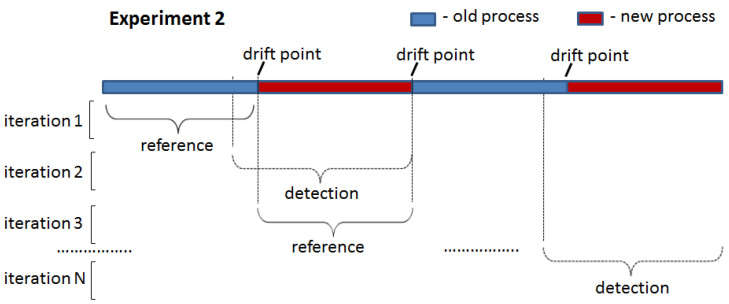
Experiment 2: drift introduced in the middle of the detection window.

**Figure 8 entropy-24-00910-f008:**
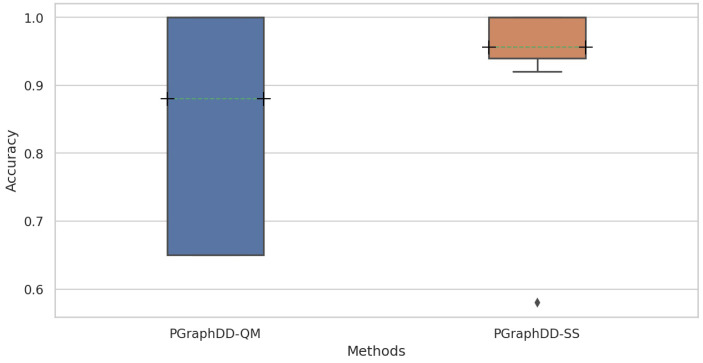
Distribution of accuracy scores achieved by PGraphDD-QM and PGraphDD-SS across all change patterns and log sizes of the Loan Application Process dataset.

**Figure 9 entropy-24-00910-f009:**
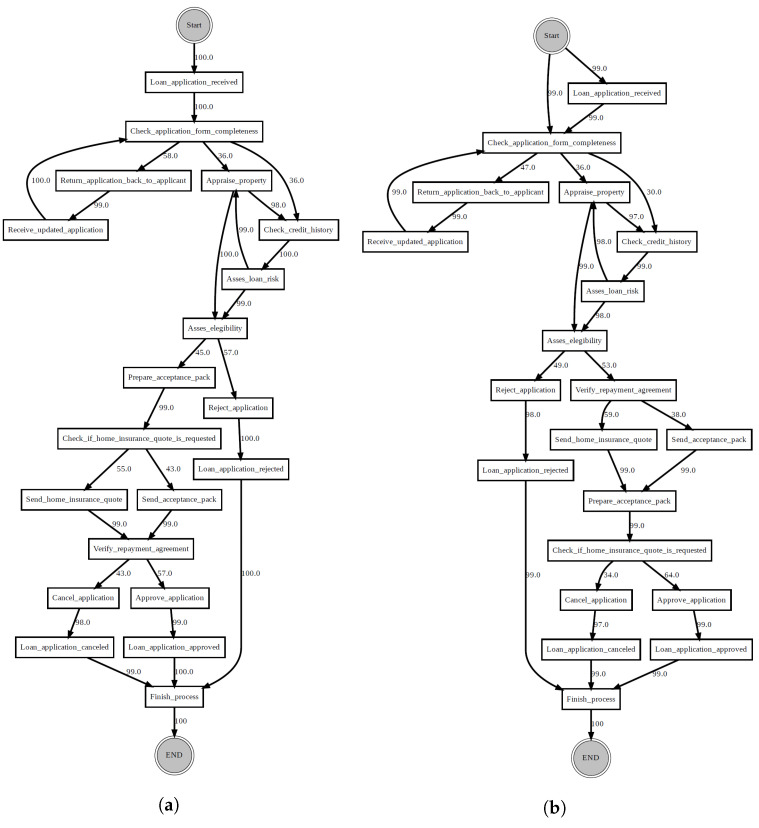
Directly follows process graphs constructed using the ‘sw’ change pattern log from the Loan Application Process dataset: (**a**) reference window; (**b**) detection window.

**Figure 10 entropy-24-00910-f010:**
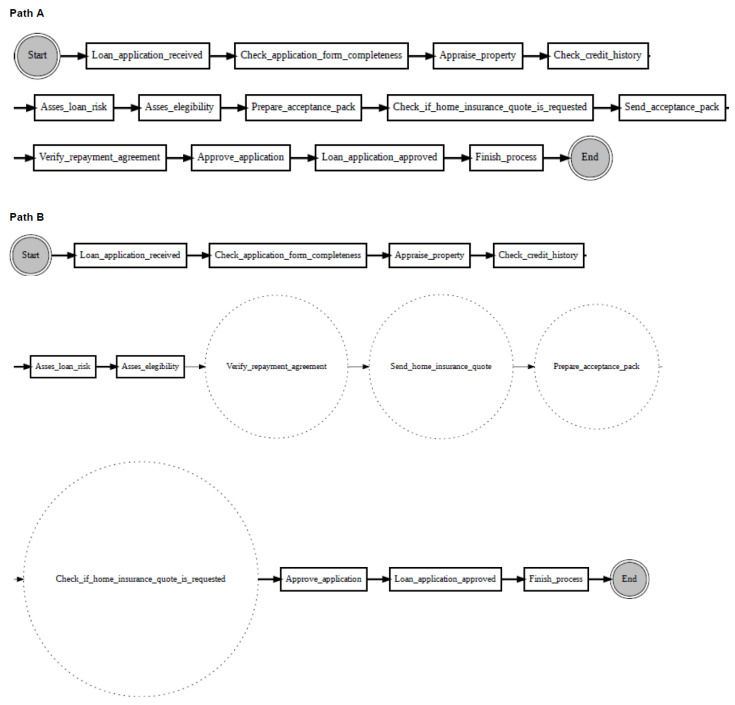
Drift localisation in the Loan Application Process logs: Path A—reference path from the reference window of the ‘sw’ process graph; Path B—candidate path from the detection window of the ‘sw’ process graph.

**Figure 11 entropy-24-00910-f011:**
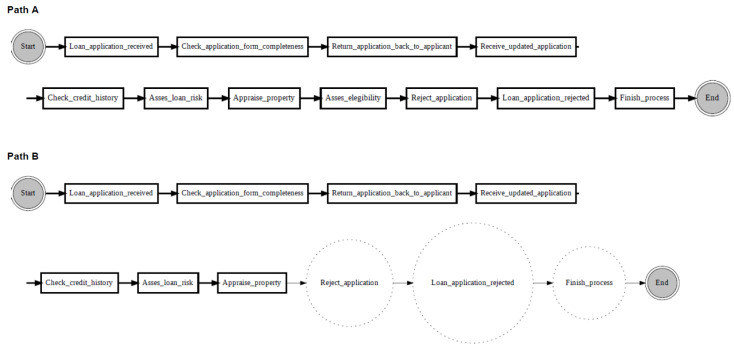
Drift localisation in the Loan Application Process logs: Path A—reference path from the reference window of the ‘re’ process graph; Path B—candidate path from the detection window of the ‘re’ process graph.

**Figure 12 entropy-24-00910-f012:**
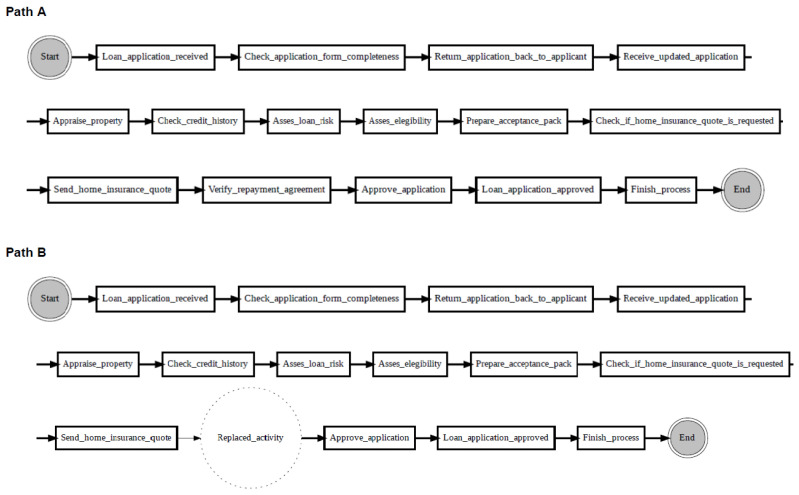
Drift localisation in the Loan Application Process logs: Path A—reference path from the reference window of the ‘rp’ process graph; Path B—candidate path from the detection window of the ‘rp’ process graph.

**Figure 13 entropy-24-00910-f013:**
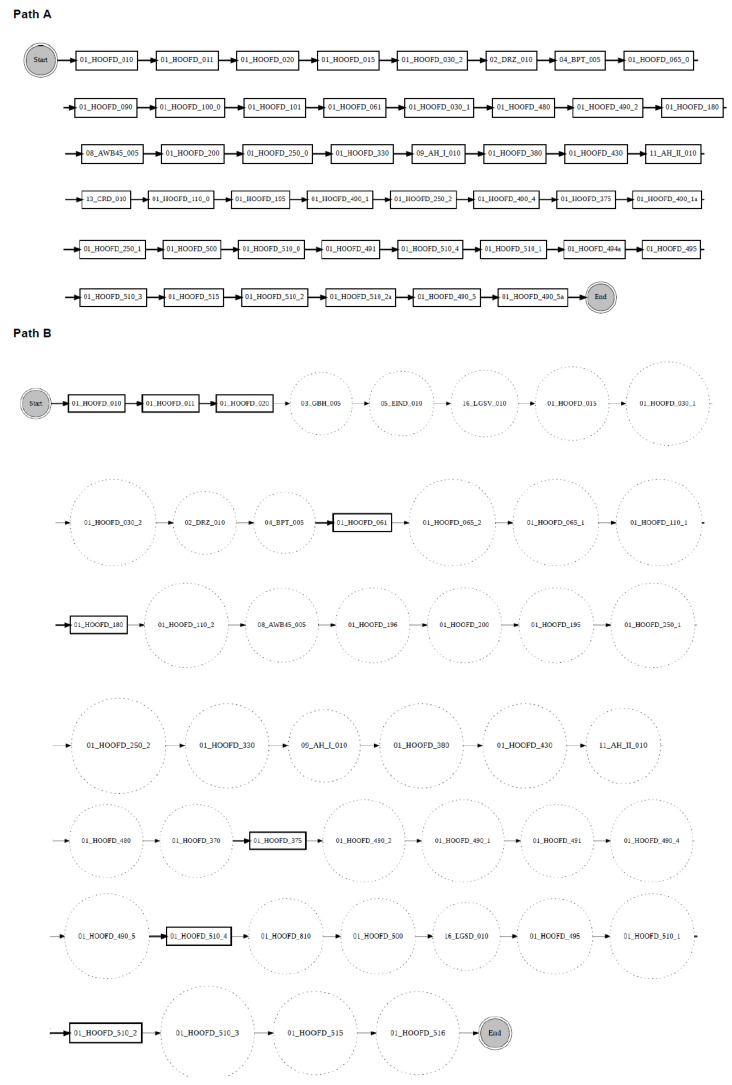
Drift localisation: Path A—reference path from the BPIC 2015 reference process graph; Path B—candidate path from the BPIC 2015 detection process graph.

**Figure 14 entropy-24-00910-f014:**
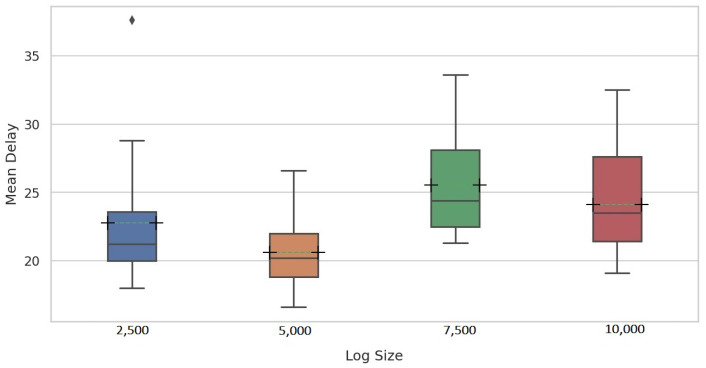
Distribution of mean delay scores achieved by PGraphDD-QM per change pattern for each of the four log sizes of the Loan Application Process dataset.

**Figure 15 entropy-24-00910-f015:**
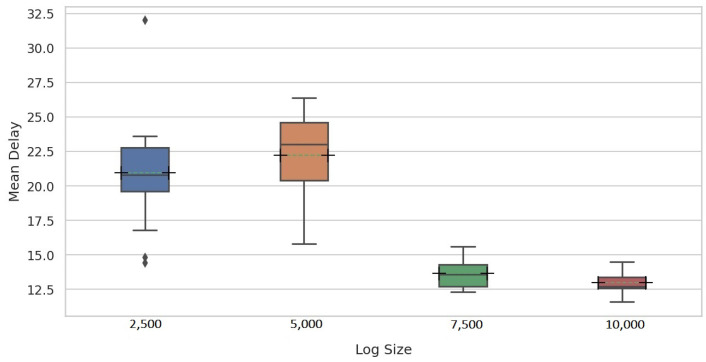
Distribution of mean delay scores achieved by PGraphDD-SS per change pattern for each of the four log sizes of the Loan Application Process dataset.

**Figure 16 entropy-24-00910-f016:**
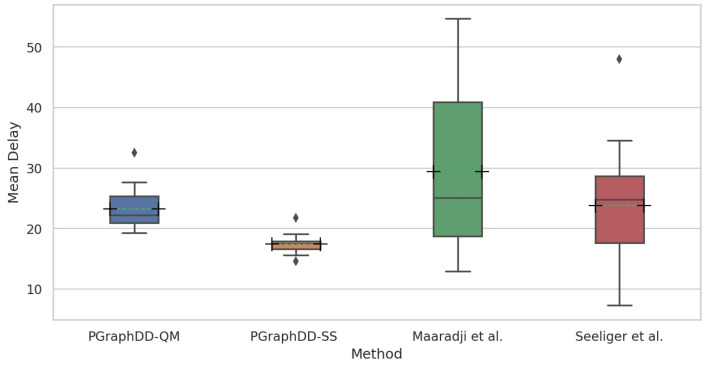
Distribution of mean delay scores per change pattern across all log sizes of the Loan Application Process dataset: PGraphDD-QM and PGraphDD-SS versus Maaradji et al. [14] and Seeliger et al. [9].

**Table 1 entropy-24-00910-t001:** Control-flow change patterns for synthetic event logs.

Code	Simple Change Pattern	Category
re	Add/remove fragment	I
cf	Make two fragments conditional/sequential	R
lp	Make fragment loopable/non-loopable	O
pl	Make two fragments parallel/sequential	R
cb	Make fragment skippable/non-skippable	O
cm	Move fragment into/out of conditional branch	I
cd	Synchronise two fragments	R
cp	Duplicate fragment	I
pm	Move fragment into/out of parallel branch	I
rp	Substitute fragment	I
sw	Swap two fragments	I

**Table 2 entropy-24-00910-t002:** Results across all log sizes for the “cd” change pattern of the Loan Application Process dataset using PGraphDD-QM.

Windows	cd-2500	cd-5000	cd-7500	cd-10000	Drift Detected?	Actual Drift?	Correct Detection?
**0,1**	0.98	0.99	0.99	0.99	No	No	Yes
**1,2**	0.53	0.47	0.55	0.50	Yes	Yes	Yes
**2,3**	0.50	0.94	0.99	0.99	No	No	Yes
**3,4**	0.62	0.49	0.54	0.47	Yes	Yes	Yes
**4,5**	0.96	0.99	0.99	0.99	No	No	Yes
**5,6**	0.42	0.45	0.50	0.49	Yes	Yes	Yes
**6,7**	0.69	0.99	0.99	0.99	No	No	Yes
**7,8**	0.56	0.48	0.48	0.63	Yes	Yes	Yes
**8,9**	0.94	0.98	0.99	0.99	No	No	Yes
**9,10**	0.89	0.51	0.49	0.49	Yes	Yes	Yes
**10,11**	0.98	0.97	0.99	0.99	No	No	Yes
**11,12**	0.25	0.50	0.48	0.48	Yes	Yes	Yes
**12,13**	0.95	1.00	0.99	0.99	No	No	Yes
**13,14**	0.37	0.49	0.44	0.49	Yes	Yes	Yes
**14,15**	0.65	0.99	0.98	0.99	No	No	Yes
**15,16**	0.50	0.45	0.52	0.48	Yes	Yes	Yes
**16,17**	0.96	0.99	0.99	0.99	No	No	Yes
**17,18**	0.82	0.45	0.48	0.51	Yes	Yes	Yes
**18,19**	0.92	0.97	1.00	0.99	No	No	Yes
**19,0**	0.33	0.46	0.67	0.49	Yes	Yes	Yes

**Table 3 entropy-24-00910-t003:** Results across all log sizes for the “cd” change pattern of the Loan Application Process dataset using PGraphDD-SS.

Windows	cd-2500	cd-5000	cd-7500	cd-10000	Drift Detected?	Actual Drift?	Correct Detection?
**0,1**	1.00	1.00	0.93	1.00	No	No	Yes
**1,2**	0.86	0.86	0.86	0.86	Yes	Yes	Yes
**2,3**	1.00	1.00	1.00	1.00	No	No	Yes
**3,4**	0.86	0.86	0.86	0.86	Yes	Yes	Yes
**4,5**	1.00	1.00	1.00	1.00	No	No	Yes
**5,6**	0.86	0.86	0.86	0.86	Yes	Yes	Yes
**6,7**	1.00	1.00	1.00	1.00	No	No	Yes
**7,8**	0.86	0.86	0.86	0.93	Yes	Yes	Yes
**8,9**	1.00	0.93	1.00	0.93	No	No	Yes
**9,10**	0.86	0.93	0.86	0.86	Yes	Yes	Yes
**10,11**	1.00	1.00	1.00	1.00	No	No	Yes
**11,12**	0.86	0.86	0.86	0.86	Yes	Yes	Yes
**12,13**	1.00	1.00	1.00	1.00	No	No	Yes
**13,14**	0.86	0.86	0.86	0.86	Yes	Yes	Yes
**14,15**	1.00	1.00	1.00	1.00	No	No	Yes
**15,16**	0.86	0.86	0.86	0.86	Yes	Yes	Yes
**16,17**	1.00	1.00	1.00	1.00	No	No	Yes
**17,18**	0.86	0.86	0.86	0.86	Yes	Yes	Yes
**18,19**	1.00	1.00	1.00	1.00	No	No	Yes
**19,0**	0.86	0.92	0.86	0.86	Yes	Yes	Yes

**Table 4 entropy-24-00910-t004:** Results obtained for BPIC 2015 using PGraphDD-QM.

Windows	F-Score	Drift Detected?	Actual Drift?	Correct Detection?
0,1	0.92	No	No	Yes
1,2	0.87	Yes	Yes	Yes
2,3	0.88	Yes	No	No
3,5	0.89	Yes	Yes	Yes
4,5	0.90	No	No	Yes
5,6	0.21	Yes	Yes	Yes
6,7	0.84	Yes	No	No
7,8	0.67	Yes	Yes	Yes

**Table 5 entropy-24-00910-t005:** Results obtained for BPIC 2015 using PGraphDD-SS.

Windows	Similarity Score	Drift Detected?	Actual Drift?	Correct Detection?
**0,1**	0.31	No	No	Yes
**1,2**	0.27	Yes	Yes	Yes
**2,3**	0.36	No	No	Yes
**3,5**	0.15	Yes	Yes	Yes
**4,5**	0.32	No	No	Yes
**5,6**	0.30	No	Yes	No
**6,7**	0.39	No	No	Yes
**7,8**	0.24	Yes	Yes	Yes

**Table 6 entropy-24-00910-t006:** Reasons for drifts extracted from graphs generated using PGraphDL for each change pattern of the Loan Application Process dataset.

Change Pattern	Reason for Drift Extracted from Graphs	Reason Correct?
**cb**	‘Prepare acceptance pack’ and ‘Check application form completeness’ are non-skippable before the drift but skippable after drift.	Correct
**cd**	‘Check credit history’ and ‘Assess loan risk’ are sequential before the drift but happen at the same time after drift.	Correct
**cf**	‘Send home insurance quote’ and ‘send acceptance pack’ are conditional before the drift but sequential after drift.	Correct
**ori**	‘Added activity’ is added in-between Send home insurance quote’ and ‘send acceptance pack’ which are made sequential and loopable after the drift.	Correct
**rp**	‘Verify repayment agreement’ is always in place before the drift but replaced with ‘Replaced activity’ after the drift.	Correct
**sw**	‘Prepare acceptance pack’ and ‘Check if home insurance quote is requested’ swapped place with ‘Verify repayment agreement’ after the drift.	Correct

**Table 7 entropy-24-00910-t007:** Average accuracy achieved over the Loan Application Process logs by the proposed and state-of-the-art methods.

#	PGraphDD-QM	PGraphDD-SS	Maaradji et al. [14]	Seeliger et al. [9]	Gaspar et al. [13]
**cb**	0.65	0.92	0.92	**0.97**	0.76
**cd**	**1.00**	0.97	0.88	0.95	0.90
**cf**	0.98	**1.00**	0.98	0.98	0.98
**cm**	0.65	0.92	**1.00**	0.97	0.91
**cp**	**1.00**	**1.00**	**1.00**	0.98	**1.00**
**ior**	0.65	0.58	**1.00**	0.96	-
**iro**	**1.00**	**1.00**	**1.00**	0.94	-
**lp**	0.73	**1.00**	**1.00**	0.76	0.76
**oir**	**1.00**	**1.00**	0.98	0.73	-
**ori**	**1.00**	0.94	**1.00**	0.98	-
**pl**	0.65	**1.00**	**1.00**	0.95	0.90
**pm**	**1.00**	**1.00**	**1.00**	0.98	1.00
**re**	**1.00**	**1.00**	**1.00**	0.90	0.72
**rio**	0.95	**1.00**	0.98	0.97	-
**roi**	**1.00**	0.92	**1.00**	**1.00**	-
**rp**	**1.00**	**1.00**	0.96	0.97	0.98
**sw**	**1.00**	**1.00**	**1.00**	**1.00**	**1.00**

**Table 8 entropy-24-00910-t008:** Mean delay per change pattern of the Loan Application Process dataset compared to Maaradji et al. [14] and Seeliger et al. [9].

#	Mean Delay *d*			
	PGraphDD-QM	PGraphDD-SS	Maaradji et al. (ADWIN) [14]	Seeliger et al. [9]
cb	25.5	**18.1**	54.7	18.9
cd	27.7	**17.8**	32.4	28.7
cf	25.4	**16.7**	19.1	34.6
cm	21.0	**17.9**	40.9	19.2
cp	24.4	**16.6**	18.7	17.6
ior	21.0	17.5	16.7	**13.0**
iro	19.7	**16.5**	43.8	27.2
lp	19.3	**18.8**	46.3	48.0
oir	19.5	**17.1**	43.7	28.01
ori	20.5	17.8	**12.9**	14.3
pl	32.5	**19.1**	36.5	26.3
pm	20.9	17.1	**12.9**	24.8
re	25.0	**14.6**	38.1	33.0
rio	22.2	**17.9**	25.1	20.8
roi	21.9	21.8	19.8	**7.3**
rp	25.5	15.6	16.9	**12.7**
sw	25.7	**15.9**	20.8	29.6

## Data Availability

The datasets used in this study, along with code, can be found as follows:Loan Application Process logs [14]: https://data.4tu.nl/articles/dataset/Business_Process_Drift/12712436.Dutch Municipality logs (BPIC 2015) [17]: https://data.4tu.nl/collections/_/5065424/1.Source code: https://github.com/dijahanga/DL_Approach_To_Process_Mining.git. Loan Application Process logs [14]: https://data.4tu.nl/articles/dataset/Business_Process_Drift/12712436. Dutch Municipality logs (BPIC 2015) [17]: https://data.4tu.nl/collections/_/5065424/1. Source code: https://github.com/dijahanga/DL_Approach_To_Process_Mining.git.

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
