# Peer review of "PGraphD*: Methods for Drift Detection and Localisation Using Deep Learning Modelling of Business Processes"

_entropy, 2022, doi:10.3390/e24070910_

Round 1

Reviewer 1 Report

This paper contains a strong algorithmic analysis of a set of methods in process mining (drift detection, drift localization). The methods are based on deep learning and graphs. The paper is well formalized and presented but it is written from a strictly algorithmic point of view and authors should make some practical considerations of their work, connecting their algorithms with the business process management concept and the software that can embed these algorithms in order to be practically applied in real-life problems. So, some improvements are necessary before it can be published.

1)      The abstract is too short and totally lacks info on the background of the research. The authors need to follow the “instructions for authors” in order to write a proper abstract.   

2)      The authors correctly use the keyword “business process management” but they do not describe it, as well as its relation to process mining in their introduction section. I am waiting to see an introduction to BPM, its definition, some references to BPM practical applications, and what is its relation to process mining. I suggest authors extend their introduction and include the following literature in order to present BPM and place the process mining into the BPM concept:

·         Reijers Hajo A. (2021) Business Process Management: The evolution of a discipline, Computers in Industry, Volume 126, April 2021, 103404, https://doi.org/10.1016/j.compind.2021.103404

·         van der Aalst, W.M.P., La Rosa, M. & Santoro, F.M. Business Process Management. Bus Inf Syst Eng 58, 1–6 (2016). https://doi.org/10.1007/s12599-015-0409-x

·         Panayiotou N.A., Stavrou V.P., Gayialis S.P. (2017) The Application of a Business Process Modeling Architecture in the Supply Chain of a Manufacturing Company: A Case Study. In: Grigoroudis E., Doumpos M. (eds) Operational Research in Business and Economics, p.p. 1-16. Springer Proceedings in Business and Economics. Springer, Cham https://doi.org/10.1007/978-3-319-33003-7_1.

·         Zuhaira, B. and Ahmad, N. (2021), "Business process modeling, implementation, analysis, and management: the case of business process management tools", Business Process Management Journal, Vol. 27 No. 1, pp. 145-183. https://doi.org/10.1108/BPMJ-06-2018-0168

·         Panayiotou, N.A., Gayialis, S.P., Evangelopoulos, N.P. and Katimertzoglou, P.K. (2015), "A business process modeling-enabled requirements engineering framework for ERP implementation", Business Process Management Journal, Vol. 21 No. 3, pp. 628-664. https://doi.org/10.1108/BPMJ-06-2014-0051

·         Park S. and Kang Y.S. (2016) A Study of Process Mining-based Business Process Innovation, Procedia Computer Science, Volume 91, 2016, Pages 734-743, https://doi.org/10.1016/j.procs.2016.07.066

·         Panayiotou, N.A. and Stergiou, K.E. (2022), "Development of a retail supply chain process reference model incorporating Lean Six Sigma initiatives", International Journal of Lean Six Sigma, Vol. ahead-of-print No. ahead-of-print. https://doi.org/10.1108/IJLSS-04-2021-0079

·         Graafmans, T., Turetken, O., Poppelaars, H. et al. Process Mining for Six Sigma. Bus Inf Syst Eng 63, 277–300 (2021). https://doi.org/10.1007/s12599-020-00649-w

3)      The use of English is good, but the text should be double-checked for coherency and grammatical or syntax errors.

4)      The formation of the figures and graphs should be also be checked to be easily readable (not so small texts). Please reassign figures in the text if necessary. Figure 11 covers the caption of Figure 10.

5)      The authors should explain how the graphs in Figure 9 were created? By an automated process of a commercial software tool or it is manually created graphs?

6)      The methods and the algorithmic analysis are of high quality but there is no discussion section analyzing the findings of the paper in a qualitative and elaborative way. Additionally, the conclusions only present a summary of the whole study without mentioning the implications of the research, its contribution to scholarship and sufficient future research goals.

7)      Authors should explain the benefits of their methods and approach in relation to established tools for process mining embedded in software; not just to compare their theoretical results with benchmarks, but with other applied tools.

8)      Can the presented methods be easily applied in software? Do authors intend to develop such a software tool? This could be a clear comment to further research.

Very good research work. I suggest the above major changes improve the paper and publish it in the Entropy journal.

Reviewer 2 Report

The motivation of the work is to identify changes (drift) in dynamic business process. In this paper the authors proposed new methods for drift detection and drift localisation in business processes. These new methods are developed based on deep learning and graphs. For experimentations the authors used both synthetic and real-life logs and showed high detection accuracy. In addition, drift can be detected in shorter delay in comparison to other methods.

The introduction is brief but it provides clear summary of research area. The authors also positioned their work in comparison to similar techniques before elaborating their literature review in Section 2. Section 3 is the main content of this work, started by formal definitions and technical background before detail proposals are explained. The experiment section is clear and the illustrations complement this section nicely. The result section describes elaborate outcomes.

Overall, the paper is of high quality and it is written following good research paper standard. The topic (drift detection) itself is quite popular and it will gain a lot of interest from the research community.

Author Response

The introduction is brief but it provides clear summary of research area. The authors also positioned their work in comparison to similar techniques before elaborating their literature review in Section 2. Section 3 is the main content of this work, started by formal definitions and technical background before detail proposals are explained. The experiment section is clear and the illustrations complement this section nicely. The result section describes elaborate outcomes.

Overall, the paper is of high quality and it is written following good research paper standard. The topic (drift detection) itself is quite popular and it will gain a lot of interest from the research community.

No issues to address.

Round 2

Reviewer 1 Report

The paper is improved. I suggest it be published. Best regards.

Author Response

Thank you very much